# Farm diversification as a potential success factor for small-scale farmers constrained by COVID-related lockdown. Contributions from a survey conducted in four European countries during the first wave of COVID-19

**Zsófia Benedek**[1]*, **Imre Fertő**[1,2], **Cristina Galamba Marreiros**[3], **Pâmela Mossmann de Aguiar**[3], **Cristina Bianca Pocol**[4], **Lukáš Čechura**[5], **Anne Põder**[6], **Piia Pääso**[6], **Zoltán Bakucs**[1,7]

1 Centre for Economic and Regional Studies, Budapest, Hungary, 2 Hungarian University of Agricultural and Life Sciences, Kaposvár, Hungary, 3 Centre for Advanced Studies in Management and Economics, University of Évora, Évora, Portugal, 4 University of Agricultural Sciences and Veterinary Medicine of Cluj Napoca, Cluj Napoca, Romania, 5 Czech University of Life Sciences, Prague, Czech Republic, 6 Estonian University of Life Sciences, Tartu, Estonia, 7 Óbuda University, Budapest, Hungary

* benedek.zsofia@krtk.hu

## Abstract

This paper explores to what extent product and marketing channel diversification contributed to the economic success of small-scale agricultural producers involved in short food supply chains after the outbreak of the COVID-19 pandemic. A survey was conducted between April and July 2020 in four countries of the European Union–Estonia, Hungary, Portugal and Romania,–resulting in a relatively large sample of farmers (N = 421). The analysis was built on a semi-nonparametric approach. Approximately 19 percent of small-scale producers were able to increase sales during the first wave of the pandemic, although country-level variation was significant. Fruits and vegetables were by far the most popular products. The importance of specific channels varied across countries, but farm gate sales were among the most important marketing channels both before and during the first wave. The importance of channels that were based on digital resources and home delivery increased. Our evidence indicates that diversification was a strategy that paid off, both in terms of marketing channels and different product categories. However, the impact appears to be nonlinear; the initial advantage generated by diversification rapidly tapered off, either temporarily (in the case of products), or permanently (in the case of marketing channels). Later research may clarify whether these findings are generalizable in other socio-economic contexts, as well as in a non-COVID situation.

## Introduction

The outbreak of the Coronavirus disease 2019 (COVID-19) pandemic caused extraordinary disruption to the global food distribution system [1–3]. To save time and to 'flatten the

**Data Availability Statement:** All relevant data are within the manuscript and its Supporting Information files.

**Funding:** This study was supported by funds from the Portuguese Foundation for Science and Technology (UIDB/04007/2020). This study was also supported by the Hungarian National Research, Development and Innovation Fund in the form of grants [IF (130485), Z. Bakucs (135387), Z. Benedek (135460)] and by the Ministry of Agriculture of the Czech Republic, Program ZEMĚ in the form of funds to LC (QK1920398).

**Competing interests:** The authors have declared that no competing interests exist.

pandemic curve', most countries imposed social restrictions–most importantly, on movement–although the severity of measures differed. Nonetheless, some general tendencies were identifiable, including the collapse of just-in-time distribution systems [1, 2, 4], and an increase in consumer demand for fresh and trustworthy local food [3, 5–7], home deliveries [8], and online shopping options [9, 10]. Lockdown measures also meant that many local farmers lost contact with their customers, thus the impact of the pandemic was far from homogenous [1]. The virus, while simultaneously causing economic and social disturbances at multiple scales [11], also created a unique possibility to implement a quasi-experiment [12] involving analysis of the reactions of supply chain stakeholders to external factors.

Small farms have been claimed for some time to contribute to food and nutrition security, not only in the Global South, but even at local and regional levels in largely industrialized regions such as Europe [13, 14], in spite of their decreasing numbers [11]. In this paper we focused on small-scale farmers who participated in a number of short food supply chains (SFSCs) in four European countries (Estonia, Hungary, Portugal and Romania) to better understand how the level of diversification, both in terms of production and marketing channels, contributed to their success during the first wave of COVID.

The remainder of the paper is structured as follows. After defining SFSCs and their connection to local and small-scale agricultural systems, the rest of the introduction focuses on the literature about the benefits of diversification, with special emphasis on SFSCs. The differences in the agri-food systems of the countries included in the study are briefly described, together with a short comparison of their different COVID-related coping strategies (lockdown measures) which may have affected sales made through SFSCs. The following section briefly discusses the survey that was implemented and makes some observations related to the quantitative methods applied therein. The impact of the diversification of products and marketing strategies on economic success is then discussed. The final section concludes.

## Linkages between short food supply chains and small-scale farmers

Consumer interest in local food remains high, even outside of the pandemic situation [15]. Despite the growing attention of researchers and policy makers to local food systems, alternative food networks and short food supply chains, their respective definitions remains unclear [16, 17]. In our study we followed the approach of Gruchmann et al. [18] and Schmutz et al. [17], mainly focusing on producer-consumer interactions involving producers directly selling their products to consumers, or through a limited number (ideally, zero) of intermediaries. The reason for this choice is methodological: the flow of products through marketing channels may be tracked more accurately this way than when a specific (and highly arbitrary) geographical range is defined. This approach is similar to the logic behind several legislative instruments [e.g., 19] that support small, local agricultural businesses through the regulation of their marketing channels.

In general, but depending on the specific channel, SFSCs do not necessarily involve spatial proximity nor local purchases [20]. However, in the pandemic situation, borders were closed, isolation increased, and the SFSCs that escaped closure were expected to deliver local food.

Direct selling is a strategy typically employed by smaller agricultural holdings, and is widespread [21, 22]. The exemption of local farmers, or farmers operating in selected marketing channels (such as farmers' markets) from regulations is a regulatory tool that is often used to support small agricultural businesses [19]. As above mentioned for short food supply chains, also defining small-scale farmers is challenging [14, 23] and it is often based on certain thresholds that are highly dependent on the geographical context of the analysis [14]. Since the countries involved in our study represent markedly different contexts [14, 24] and building on the

considerations of Kneafsey et al. [21] and Martinez et al. [22], small-scale farmers were identi-fied through their participation in short food supply chains, instead of using a specific thresh-old. The terms 'small-scale farmers', 'farmers participating in SFSCs', and 'local farmers' are used interchangeably in this paper.

## Diversification from the perspective of small-scale farmers

Farm diversification is often analysed in the context of multifunctional agriculture, when the potential of agricultural enterprises to produce products and services other than food and fibre is discussed [25]. Salvioni et al. [26] proposed a classification that distinguishes between off- and on-farm diversification. Off-farm diversification is achieved through 'pluriactivity', when farm household members increase their income from sectors other than agriculture [27]. According to Salvioni et al. [26], on-farm diversification includes three broad areas: a) agricul-tural output diversification (e.g. selling a specific mix of products); b) product differentiation (e.g. the production of organic, or products of protected designation of origin); and, c) non-agricultural output diversification (e.g. involvement in agritourism, natural resource manage-ment, etc.). [11] found that agricultural output diversification (and the combination of on-farm agricultural and non-agricultural activities, or on-farm and off-farm work) is a typical resilience strategy of small European farms. The actual type of product [24] as well as the level of output diversification [28] have been found to determine the commercialization strategies of European small-scale farmers to some extent (e.g. products requiring processing are more typically marketed through vertically integrated supply chains; or the market linkages and the actual choice of short food supply chain marketing channels differ among diversified and more focused businesses). However, the conceptualization of and knowledge about the diversi-fication of marketing channels is very limited, although it is acknowledged that tracking the use of channels is needed for a more complete understanding (and thus the efficient support) of local and regional food systems [29]. In this paper we focused on agricultural output diversi-fication in the sense that farmers are regarded as diversified if they worked with more than one product category (e.g. fruits and vegetables, milk and dairy, meat, honey, etc.). Additionally, diversification is identified when a farmer makes sales through more than one direct market-ing channel.

Increasing diversification is a key element of European Union (EU) Common Agricultural Policy, as well as of Horizon 2020 strategies [7, 30]. However, in EU terminology, and also in a number of studies, off-farm and/or non-agricultural diversification are often pursued or ana-lysed; there are thus considerable knowledge gaps related to diversification concerning core agricultural activities and marketing channel use.

The reasons that farmers diversify their businesses are rather complex [31]. Diversification is an important element of risk management under uncertainty [7, 32, 33], and the adoption of on- and off-farm diversification strategies is often an element of risk minimization [34]. A fur-ther reason is the presence of economies of scope [35, 36]. According to Panzar and Willing [37], economies of scope are generated when inputs are shared, and the cost of producing two or more product lines is less than the cost of producing each line separately. A business that takes full advantage of economies of scope produces complementary items (classic examples include eggs and poultry, mutton and wool, milk and meat, etc.), so a fuller range of products can sent to market [32]. Additionally, diversification and joint production can help manage peaks in labour demand [38] and to fully exploit other kinds of resources efficiently [39], thus increase productivity. Finally, diversification may be a means of increasing sustainability, mostly from an agroeconomical and social perspective [38, 40]. The diversification of market-ing channels can create advantages such as fostering access to different markets and price

premiums, mitigating marketing risks, utilizing different niches, and the better management of the seasonality of agricultural production and variability in product quality [41].

Due to the small size of the farms that are the focus of this paper, the question may be raised whether these businesses can benefit from specialization, technological efficiency, and economies of scales at all, or whether diversification, resilience, and economies of scope would be more promising strategies instead. The choice of specialization versus diversification is reportedly significant according to farm size. Large and very large farms tend to specialize and intensify more to achieve economies of scale [42], while diversification is a typical strategy of smaller businesses [32]. The tendency among organic farms is similar, while as with conventional businesses, larger and more specialized farms tend to send products to market through long (global) supply chains [43]. However, several exceptions to this have been found. For example, larger farms are more diversified in Norway [44], Austria [45] and the Netherlands [46], while small rice farms are highly specialized in Korea [36], due to local social, political, economic, and environmental characteristics. Moreover, some studies suggest that diversification might increase with farm size on the basis that bigger farms can allocate and exploit resources more efficiently [47, 48]. To conclude, the effect of farm size on the level of specialization/diversification is unclear both theoretically and empirically. Thus, no prior assumptions were made in relation to the analysis; the level of diversification among small-scale farms was assumed to be heterogeneous.

## Materials and methods

To increase the relevance and robustness of the results, data from countries with very different characteristics were used in the analysis. The sample includes a Mediterranean country (Portugal), a Baltic state (Estonia), and two Central and Eastern European countries (Hungary and Romania). In addition, the first wave of the COVID-19 crisis hit these economies differently, prompting responses ranging from relatively mild restrictions (e.g. Hungary) to very strict ones (e.g. Portugal). Statistical data that characterize the target-countries of this paper were explored, and their differences and specificities were discussed. Farm size were measured using two alternative measurements, Economic Size (1000 EUR of Standard Output) and Total Labour used (Annual Work Unit). Standard Output is a region- and product-specific monetary value of production measured as a five-year average, at the farm-gate price. Annual Work Unit is the amount of labour of a single, full-time farm employee. Percentile (quantile) shares are commonly used to emphasise inequality, similarly to Lorenz curves. The European Commission's Farm Accountancy Data Network (FADN) data, designed to be representative, is used to depict the distribution of farm size. FADN data cover about 50% of all agricultural households in Estonia. Also, larger farms are slightly overrepresented in the FADN sample. However, the general trends identified by the data remain valid.

Although FADN data is representative, it ignores truly small farms, similarly to agricultural censuses and other national statistics [14]. Additionally, to address our specific research questions, we needed the contribution of a high number of small-scale farmers. Therefore, a survey was constructed, and data were collected through structured telephone interviews between April and July. To increase the sample size as much as possible in a situation when the willingness of farmers to participate in research was very low due to distress caused by the uncertainties of the very first wave of COVID, and the need for data collection that ensured social distancing, farmers were approached by representatives of specified networks who had already been in contact with the related farmers for a long time. All the networks focused on small-scale farmers and local food systems; they were consumer purchase groups, non-governmental organizations, Local Action Groups of the LEADER program, etc. (The LEADER program is a

European Union initiative to support rural development projects launched at the local level through Local Action Groups in order to revitalise rural areas and create jobs. The term 'LEADER' comes from the French acronym for "Liaison Entre Actions de Développement de l'Économie Rurale", meaning 'Links between the rural economy and development actions'. The program concerns mostly small and medium-sized enterprises [49, 50], and many local food initiatives [21]. In spite of national differences, Local Action Groups are very important actors in local foodscapes by organizing the actions of small-scale farmers [17, 51]). Thus, for convenience producers were considered small-scale farmers if they were connected to these organizations, although this decision necessarily increased the limitation of our results. Informed verbal consent was obtained from all the respondents.

The Ethics Review Procedure complied with the Guidance Note "Ethics in Social Sciences and Humanities", issued by the European Commission in 2018. The questionnaire, the use of verbal consent and other ethical aspects were approved by the Ethical Committee of the Centre for Economic and Regional Studies, Hungary–the lead country of this research. The questionnaire was translated and used to collect data in other countries (all of them part of the European Union). The variables used in the analysis are described below.

Changes in sales due to COVID-related restrictions–our dependent variable–is described by a binary variable showing whether a producer experienced an increase in their sales (1), or not (0).

Concerning the product categories commercialized by each farmer, data were coded into dummies: "1" if yes, "0" otherwise. In total, eight sectors were considered: fruits and vegetables, egg or poultry, meat, milk and dairy, bakery products, herbs and spices, honey, and grapes or wine. A measure of 'product diversification' was created *a posteriori* that was equal to the number of different product categories respondents mentioned using.

Producers also evaluated the importance of the direct marketing channels they used, both before and during the first wave of COVID in terms of the income these channels generated. A five-point Likert scale was employed (1: I do not use this channel; 5: this marketing channel is very important for my sales). To characterize the level of diversification, the number of channels producers used both before and during COVID was calculated *a posteriori*. Only important channels were involved in the estimations. Thus, the variable 'channel diversification' shows how many marketing channels a given producer deemed important (i.e. those that were assigned a level of importance of four or five on the five-item Likert scale).

Finally, producers were requested to identify their annual gross farm income (off-farm activities included, without government farm program payments), in line with one of the predefined categories: 1: below 5,000 Euro (EUR); 2: 5,000–15,000 EUR; 3: 15,000–30,000 EUR; 4: 30,000–50,000 EUR; 5: more than 50,000 EUR.

Four binary variables were created *a posteriori* for all the countries to express geographical location to control for country-level differences. Romania was chosen as the benchmark, as the Romanian subsample is the largest.

Twelve short food chain marketing channels were included in the survey: markets, farm gate sales, festivals, farmstay and farmland food-related services, restaurants, purchasing groups, independent shops, retail chains, public procurement, home delivery, own ecommerce store, and directory (sales made through the website of a fellow producer or an association). To reduce the number of variables in the final calculations, factor analysis with the principal component factor method was undertaken to capture the differences in the importance of channels before and after the outbreak of COVID. First, the data were tested to determine the applicability of the method, using Kaiser–Meyer–Olkin's measure and Bartlett's test of variable independence, followed by the Varimax rotation algorithm. Finally, Kaiser selection criteria

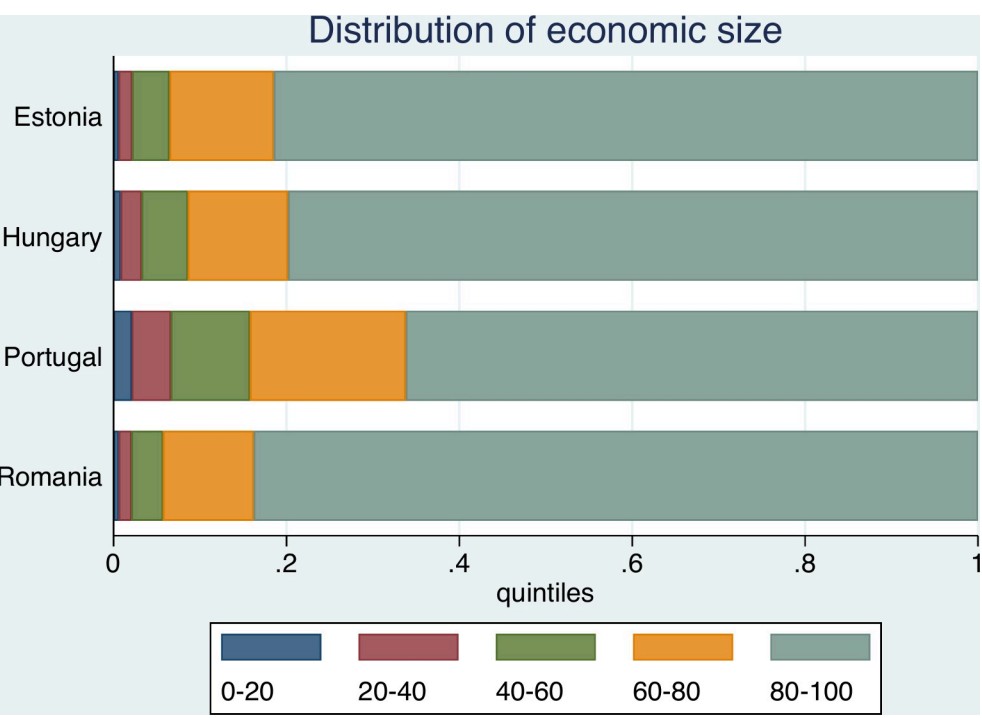

**Fig 1. Distribution of economic size of farms in sample countries.** Source: FADN.

were applied, considering only factors with Eigenvalues exceeding one (see [52] for the practitioner's handbook). The resulting factors were used as explanatory variables in the models.

Since our dependent variable is binary ("1" if sales increased during COVID and "0" otherwise), the impact of diversification and other variables on whether a producer experienced success during COVID required estimation using binary choice models. We used the semi-nonparametric (SNP) method defined by Gallant and Nychka [53] due to its robustness when compared to the standard models. The benefit of a non-parametric approach is that, unlike parametric estimators, it is not sensitive to distributional assumptions; semi-non-parametric estimators have been reported to outperform parametric ones [54]. Robust variance-covariance matrix estimation was used to account for potential heteroscedasticity. Sequential Likelihood-ratio (LR) tests led to the choice of order 3 for the univariate Hermite polynomial expression used in the semi-nonparametric estimator.

To control for non-linear effects, the second and third power of channel and product diversities, and the second power of income were also incorporated into the estimations. Stata 16 was used for the statistical analyses.

## Results and discussion

Figs 1 and 2 illustrate the farm structures in the countries under analysis.

The pronounced unequal distribution of farms may be observed, especially for Hungary, Romania, and Estonia, where the top 20 percent of farms (grey quintile) account for more than 80% of cumulative economic farm size. Except for Portugal, the upper two quantiles contribute more than 90% of cumulative economic size. By contrast, the smallest 40% (red and blue coloured quintiles) of farms account for less than 3% of cumulative output (except for Portugal, where the figure amounts to 6.5%). Both Figs 1 and 2 thus reflect the dominance of large farms.

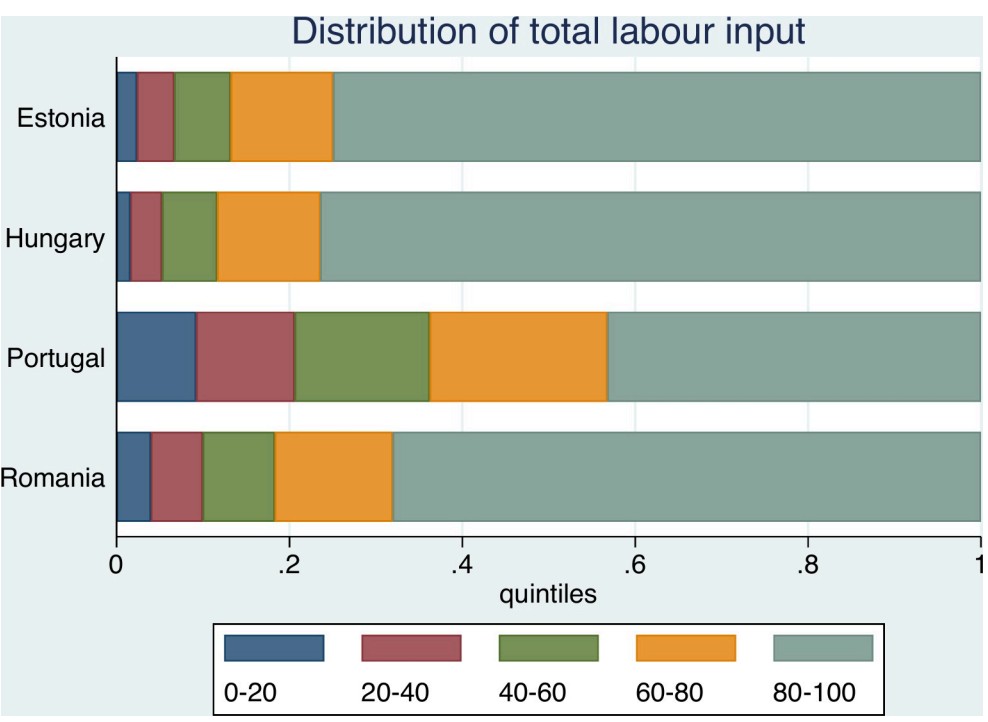

**Fig 2. Distribution of Total Labour Used by the farms in the sample countries.** Source: FADN.

Turning to the distribution of farms according to labour used (Fig 2), the picture is somewhat more balanced. Portuguese farm structure is again the most balanced, while the highest levels of inequality are registered for Hungary and Estonia. It is evident that the agricultural production structures of the former socialist economies are usually different to those found in Portugal. Whilst countries in the former region–as a legacy of socialist cooperatives and large state farms structure–typically rely on large holdings, Portuguese agricultural structure is more family farm based, resulting in a more balanced development. Guiomar et al. [14] observed that whilst in Hungary 10% of the largest farms use 80% of the total agricultural land, most of Europe's small farms are located in the South-Eastern Europe and the Mediterranean countries. Central and Northern Portugal are known for having especially small farms. It follows, that the average farm size, based on representative FADN data (Table 1) is also rather different.

The largest mean cereal farm size can be found in Estonia, about 20 times larger than the Portuguese value, the lowest one, according to the data from FADN. The distribution is somewhat more balanced for animal husbandry farms, where the average difference between the largest (Hungary) and smallest (Portugal) is five-fold. With respect to the average size of other field crops farms in the FADN sample, the post-socialist countries are rather similar, whilst Portuguese farms are the smallest.

The impact of the first wave of COVID on the population and corresponding restrictions and measures are now introduced (Table 2).

Except perhaps for Portugal, the coronavirus pandemic affected the sample countries relatively lightly, based on the number of deaths per million inhabitants until June 31 (higher figures were recorded for the United Kingdom (642), Spain (606), Italy (575), Sweden (526), and the United States of America (381)). Several factors might have contributed to this difference,

Table 1. Average farm size by some specific agricultural sectors.

| | N | Mean | SD | Min | Max |
|---|---|---|---|---|---|
| Cereals (ha) | | | | | |
| Estonia | 430 | 200.43 | 316.41 | 0.10 | 2593.10 |
| Hungary | 1508 | 119.03 | 294.79 | 0.30 | 6368.90 |
| Portugal | 519 | 10.47 | 29.71 | 0.01 | 381.00 |
| Romania | 3582 | 101.19 | 262.41 | 0.02 | 5000.00 |
| Other field crops (ha) | | | | | |
| Estonia | 431 | 63.61 | 111.28 | 0.01 | 1034.30 |
| Hungary | 1024 | 79.19 | 145.34 | 0.01 | 1867.90 |
| Portugal | 679 | 1.42 | 12.33 | 0.01 | 302.00 |
| Romania | 2318 | 79.59 | 182.42 | 0.01 | 2280.00 |
| Animal husbandry (total livestock units) | | | | | |
| Estonia | 409 | 212.56 | 428.86 | 0.28 | 2956.99 |
| Hungary | 871 | 224.67 | 1006.66 | 0.01 | 21792.99 |
| Portugal | 1171 | 42.95 | 171.52 | 0.01 | 5299.20 |
| Romania | 2270 | 66.16 | 411.80 | 0.00 | 13159.74 |

Source: FADN

including age distribution, early social distancing policies, the BCG (Bacillus Calmette–Guérin) vaccination, social psychological factors, the status of public health and social care systems, etc. [55–58]. Nevertheless, several gaps in understanding the impact of the pandemic still remain. As the focus of this paper lies elsewhere, potential explanatory variables are not analysed here: the purpose of this section is only to document the differences in mortality and some other measures that might have affected direct sales in the focal countries.

The sample countries reacted fairly similarly in terms of the implementation of COVID-related measures. Though gatherings and public events were cancelled, movement remained relatively free for the purpose of work, purchasing items to fulfil basic needs, taking care of others, and recreational activity. Telework was centrally mandated for some professionals, such as civil servants in Portugal, but many more decided to stay at home in all countries, either to supervise their children (as the institutions of education had closed everywhere), making the use of the home office as a general rule. Yet border checks were reinstalled, movement within all countries was possible, except for some selected regions such as some islands in Estonia. While shopping centers completely closed in Portugal, the stores of key service providers and markets located within shopping malls remained open in Estonia, Hungary, and Romania. Some countries introduced a shopping time window for senior citizens (Hungary, and Romania), while others advised vulnerable people to stay at home completely (Estonia, Portugal).

Table 2. COVID-19 in numbers (first wave), and related measures.

| | Estonia | Hungary | Portugal | Romania |
|---|---|---|---|---|
| Population (midyear data of the United Nations, UN) (million) | 1.327 | 9.660 | 10.197 | 19.238 |
| Population density (midyear UN data) (per km$^2$) | 31 | 104 | 111 | 84 |
| Outbreak of COVID (first confirmed case) | February 27 | March 4 | March 2 | February 25 |
| COVID death toll until June 30, 2020 (World Health Organization data) | 69 | 585 | 1568 | 1634 |
| Death toll per million people until June 30, 2020 | 52 | 61 | 154 | 85 |
| Date of lifting of most restrictions | By May 18 | By May 18 | By May 18 | By June 1 |

**Table 3. Descriptive statistics.**

| Variable | N | Frequency (%) | Mean | SD | Min | Max |
|---|---|---|---|---|---|---|
| Increase in sales | 421 | - | 0.258 | 0.438 | 0 | 1 |
| Channel diversification | 421 | - | 3.517 | 2.979 | 0 | 12 |
| Number of channels: 0 | 41 | 9.7 | - | - | - | - |
| Number of channels: 1 | 77 | 18.3 | - | - | - | - |
| Number of channels: 2 | 86 | 20.4 | - | - | - | - |
| Number of channels: 3 | 57 | 13.5 | - | - | - | - |
| Number of channels: 4–6 | 94 | 22.3 | - | - | - | - |
| Number of channels: 7 or more | 66 | 15.8 | - | - | - | - |
| Product diversification | 421 | - | 1.429 | 0.962 | 1 | 9 |
| Number of products: 1 | 308 | 73.2 | - | - | - | - |
| Number of products: 2 | 76 | 18.1 | - | - | - | - |
| Number of products: 3 | 25 | 5.9 | - | - | - | - |
| Number of products: 4 | 5 | 1.2 | - | - | - | - |
| Number of products: 5 or more | 7 | 1.6 | - | - | - | - |
| Income | 401 | - | 2.890 | 1.457 | 1 | 5 |
| Below €5,000 | 80 | 19.9 | - | - | - | - |
| €5,000 - €15,000 | 116 | 28.9 | - | - | - | - |
| €15,000 - €30,000 | 66 | 16.4 | - | - | - | - |
| €30,000 - €50,000 | 46 | 11.4 | - | - | - | - |
| More than €50,000 | 93 | 23.4 | - | - | - | - |
| Fruits and vegetables | 421 | - | 0.349 | 0.477 | 0 | 1 |
| Egg or poultry | 421 | - | 0.106 | 0.309 | 0 | 1 |
| Meat | 421 | - | 0.154 | 0.361 | 0 | 1 |
| Milk and dairy | 421 | - | 0.220 | 0.415 | 0 | 1 |
| Honey | 421 | - | 0.111 | 0.315 | 0 | 1 |
| Bakery products | 421 | - | 0.067 | 0.249 | 0 | 1 |
| Herbs | 421 | - | 0.078 | 0.269 | 0 | 1 |
| Wine and grapes | 421 | - | 0.097 | 0.297 | 0 | 1 |

Many of the regulations directly impacted SFSCs. Restaurants, bars, and cafés closed everywhere and only take-away and delivery services were allowed, but many establishments decided to close completely until restrictions were lifted. The ban on public gatherings impacted festivals significantly. Markets adopted a wide range of strategies depending on their management. Some tried to comply with the increasingly rigorous regulations, some closed, and others were converted into online operations. The elderly time window, where it was introduced, overlapped with the typical opening hours of farmers' markets, which thus witnessed a significant decrease in turnover. With the growth in social isolation, the demand for online food purchases and home delivery services increased significantly across all countries.

After discussing the specificities of the countries involved in the study, the survey sample is briefly presented. It included 421 observations (52 from Estonia; 136 from Hungary, 76 from Portugal; and 157 from Romania). Though the resulting sample is not representative (which is typically the case regarding research into small-scale farmers and short food supply chains), it is exceptionally large, and this allows for the addressing of the research questions through quantitative analysis. Descriptive statistics are displayed in Table 3.

Nearly twenty-six percent of all small-scale producers increased their sales during the first wave of the COVID pandemic. However, considerable differences exist among the sample countries in this respect–from 9.6 percent in Hungary to 40.1 percent in Romania–,

underlining the need for country-specific dichotomous explanatory variables in the estimations to control for country-fixed effects. The typical farmer in our sample used three to four marketing channels on average, and sold two different product categories. Country-specific differences are displayed in Figs 3 and 4; see also S1–S4 Tables. In line with the general conclusions of previous studies [32, 43], small-scale producers tend to sell to individual consumers directly, whilst only 41 farmers in our sample (9.7%) preferred longer distribution chains. Only those product categories are displayed in Table 3 the importance of which exceeded ten percent across the whole sample. Fruits and vegetables were by far the most popular sales items in our sample (the increased consumer interest in vegetables were also confirmed by [9, 59, 60]), while meat products were also in demand (similarly to the findings of [59]).

Fig 3 shows the number of producers who reported a specific number of channels to be important in the pre-COVID period. This ranged between 0 (long distribution chains preferred) and 12 (all focal direct channels extensively used), by country.

The availability of market linkages in the context of small farms (i.e. the degree of direct sales) is an important factor in food availability in European regions [24], even outside the COVID-related situation when borders are more open. Our results implied that in most countries, producers involved in direct marketing typically used 1–2 or 1–3 channels before the outbreak of the pandemic (this magnitude corresponds to the findings of [61]); the case of Romania stands out due to the higher number of channels that were actively used by many. Romania is generally considered as a "hotspot" of small farms [14], and the abundance of small-scale farmers apparently resulted in a heterogeneity and diversity of survival strategies, too. This claim is in line with the findings of [11], who underpinned the role of diverse resilience strategies of small-scale farmers. We found that the importance of specific channels varied across countries, but farm gate sales were ranked in the top two options before the pandemic in all the countries in the sample.

Fig 4 illustrates product diversification by showing the number of producers that were engaged with a specific number of product categories.

Product diversification seems to be more homogenous across the countries than channel diversification; most producers were involved with two different product categories. The level of product diversification was again most variable in Romania.

Next, the results of the factor analysis with principal component method (designed to reduce the 12 variables measuring the change in the importance of marketing channels) are

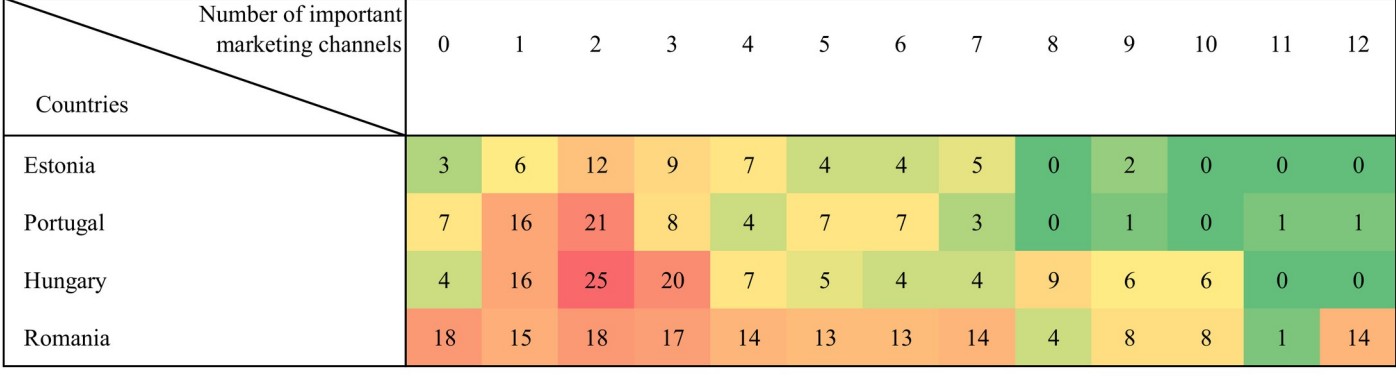

| Countries \ Number of important marketing channels | 0 | 1 | 2 | 3 | 4 | 5 | 6 | 7 | 8 | 9 | 10 | 11 | 12 |
|---|---|---|---|---|---|---|---|---|---|---|---|---|---|
| Estonia | 3 | 6 | 12 | 9 | 7 | 4 | 4 | 5 | 0 | 2 | 0 | 0 | 0 |
| Portugal | 7 | 16 | 21 | 8 | 4 | 7 | 7 | 3 | 0 | 1 | 0 | 1 | 1 |
| Hungary | 4 | 16 | 25 | 20 | 7 | 5 | 4 | 4 | 9 | 6 | 6 | 0 | 0 |
| Romania | 18 | 15 | 18 | 17 | 14 | 13 | 13 | 14 | 4 | 8 | 8 | 1 | 14 |

Number of producers

0                                                                                               25

**Fig 3. Channel diversification pre-COVID, by country.**

| Countries / Number of product categories | 1 | 2 | 3 | 4 | 5 | 7 | 8 | 9 |
|---|---|---|---|---|---|---|---|---|
| Estonia | 38 | 11 | 3 | 0 | 0 | 0 | 0 | 0 |
| Portugal | 62 | 8 | 5 | 0 | 1 | 0 | 0 | 0 |
| Hungary | 102 | 28 | 5 | 0 | 1 | 0 | 0 | 0 |
| Romania | 106 | 29 | 12 | 5 | 1 | 1 | 2 | 1 |

Number of producers

0                                                                                                   106

**Fig 4. Product diversification by country.**

briefly summarized. Three factors are identified as being in line with the minimum Eigenvalue criterion of one (see Table 4).

Factors 1 and 3 refer to channels for which the reported importance declined (many of the channels completely ceased operating). The difference is that markets, farm gate sales, and festivals were among the most important channels pre-COVID. In spite of a drop in importance, farm gate sales remained the most important mode of sales during the first wave. Factor 2 encompasses channels that became more important during COVID, thus the related change in importance is positive. These include channels based on online purchases (e.g. e-commerce stores) and home delivery was facilitated by information and communication technologies.

Table 5 presents our main results–i.e. the determinants of producer success during the outbreak of COVID, estimated by SNP models. The dependent variable, economic success during

**Table 4. Rotated factor loadings (pattern matrix).**

| Changes in importance | Factor 1 | Factor 2 | Factor 3 |
|---|---|---|---|
| Δ markets | -0.150 | -0.056 | **0.503** |
| Δ farm gate sales | -0.102 | 0.059 | **0.382** |
| Δ festivals | 0.044 | -0.084 | **0.342** |
| Δ farmstay and farmland food-related services | 0.081 | -0.065 | 0.210 |
| Δ restaurants | **0.296** | -0.087 | -0.005 |
| Δ purchase groups | **0.251** | 0.126 | -0.073 |
| Δ independent shops | **0.289** | 0.006 | -0.037 |
| Δ retail chains | **0.326** | 0.017 | -0.161 |
| Δ public procurement | **0.271** | 0.028 | -0.078 |
| Δ home delivery | -0.137 | **0.316** | 0.192 |
| Δ own ecommerce store | -0.022 | **0.459** | -0.079 |
| Δ directory | 0.095 | **0.421** | -0.124 |

Note: Numbers in bold highlight variables with correlation coefficients higher than 0.50.

**Table 5. Estimation results: The impact of selected variables on economic success.**

|  | Model 1 | Model 2 | Model 3 |
|---|---|---|---|
| Channel diversification | 1.118*** |  | 1.127*** |
| Channel diversification squared | -0.178*** |  | -0.177*** |
| Channel diversification cubed | 0.006* |  | 0.006*** |
| Product diversification |  | 4.830*** | 4.135*** |
| Product diversification squared |  | -2.302*** | -1.864*** |
| Product diversification cubed |  | 0.295*** | 0.230*** |
| Income | 0.190 | 0.050 | 0.190 |
| Income squared | -0.066*** | -0.020 | -0.066*** |
| Estonia | -1.168*** | -1.301*** | -1.403*** |
| Hungary | -2.552*** | -2.414*** | -2.881*** |
| Portugal | -1.858*** | -1.629*** | -1.975*** |
| Fruits and vegetables | 0.200 | 1.118** | 0.520 |
| Egg or poultry | 1.000*** | 1.385*** | 1.032*** |
| Meat | -0.300 | -1.450*** | -0.769** |
| Milk and dairy | 0.440 | 0.220 | 0.110 |
| Honey | -0.380 | -1.341*** | -0.970*** |
| Bakery products | 1.715*** | 0.080 | 1.170** |
| Herbs | -1.298** | -0.610 | -1.440* |
| Wine and grapes | 0.150 | 4.830*** | 0.269** |
| Factor 1 | -0.090 |  | -0.060 |
| Factor 2 | 0.445*** |  | 0.389*** |
| Factor 3 | 0.190 |  | 1.127*** |
| N | 338 | 397 | 338 |
| Wald Chi$^2$ | 372.10*** | 929.08*** | 2521.57*** |
| Log likelihood | -159.95 | -187.69 | -154.79 |
| Chi$^2$(2) LR test of Probit against SNP | 8.11*** | 11.78*** | 13.47*** |

Note

*, **, ***: significant at 10%, 5%, 1%, respectively.

the pandemic, is a binary variable that took a value of one if a farm increased its income, and was zero otherwise. Three models were estimated. The first focused on the channel diversification variables; the second targeted the variables characterizing product diversity; while the third model incorporated all the variables.

Channel diversification is very significant in all the models. Its positive sign, together with the negative sign of the quadratic and the positive sign of the cubic term, implies that the impact of marketing channel diversification on success is not linear. For Model 3, solving the following function:

$$f(x) = 1.127x - 0.177x^2 + 0.006x^3$$

(with coefficients taken from the full model) meant that the turning points could be calculated (Fig 5A).

Regarding channel diversification, the first turning point was at 3.995. Between this point and 9.29, the impact is still positive, although the marginal effect decreases. The other turning point, 15.671, is out of our scope as the maximum number of short food marketing channels is 12. In other words, diversification increases success, but only up to a specific level. Additionally, the marginal effect declines after a point. The mean number of channels used in the

overall sample was very close to this calculated optimum, with a skewed distribution according to country. Moreover, very few producers diversified their channel use beyond nine different channels (see the heat map in Fig 3). In other words, most (more than 95 percent) of the producers remained within these theoretical boundaries. The interpretation of the results based on Model 1 is very similar, and emphasises the same trends.

The impact of product diversification was not linear either. Solving the function:

$$f(x) = 4.135x - 1.864x^2 + 0.230x^3$$

(see Fig 5B) for Model 3 identified the turning points 1.559 and 3.844, with a still positive but slightly decreasing marginal effect between 1.59 and around 4. At above 4, the marginal effect steeply increases. The variables controlling for product diversification yield the same results for Model 2. To conclude, an intermediate level of product diversification increased the risk of economic loss, and economic gains were greater when product diversification was either moderate or high.

It is challenging to evaluate these results concerning the nonlinear effects of diversification in the light of previous findings that implied that increasing diversification is key to the resilience and economic success of small European farms [11, 28]. To the best of our knowledge, no prior study has explicitly analysed the impact of diversification on the performance of agricultural enterprises, and especially not in the case of farmers selling through SFSCs. However, the curvilinear effect of product diversification has been found with respect to industrial multinational enterprises [62, 63]. There is also some evidence that diversification has similarly nonlinear effects on the profitability of internationally active banks [64] and internationally diversifying restaurants [65], mainly when unrelated diversification is involved [66]. Explanations may be derived from transaction cost economic theory, which emphasizes that the need for coordination increases [67], together with bureaucratic burdens [68], and certain types of risks [66], particularly when less well-known businesses or unrelated products are involved (i.e. costs outweigh benefits beyond a certain level of diversification). However, all the cited sources emphasize that the literature that investigates linear vs. nonlinear effects includes mixed findings in the various fields of analysis–accordingly, further research is needed to confirm (challenge) the current findings with respect to the impacts of COVID, and among non-pandemic circumstances.

Farm income also impacts economic outcome; that is, whether a producer experienced success it also depended on the economic size of their operation. Second-order coefficients proved significant and negative, suggesting that larger farms might be less flexible or responsive to immediate changes in the economic environment.

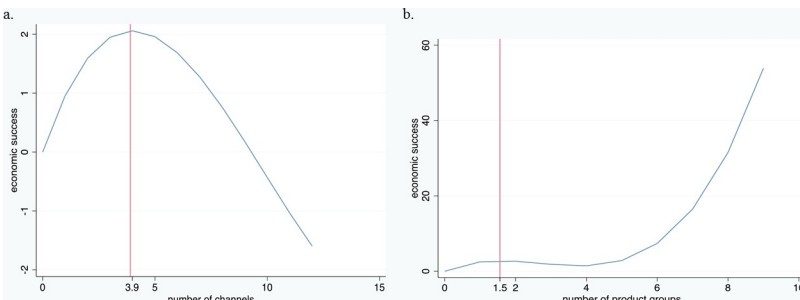

**Fig 5. Impact of diversification on economic success.** Note: a: channel diversification. b: product diversification.

As for product categories, milk and dairy did not prove to be significant in any of the models. The sign for the rest of the product categories was consistent across the models. The production of meat products had a negative impact on success. One explanation for this is that new hygiene-related demands might have arisen [69] that affected this specific product group the most. Also, fresh meat products, which are perishable, require a cool environment and large space during transportation, which is more costly and difficult to ensure in the case of home deliveries and deliveries to pick-up points. It is likely that, during the first wave of COVID, entrepreneurs were not able to access such infrastructure in time. Similar problems potentially affecting poultry might be masked by the combination of a common 'egg or poultry' category. In contrast, small-producers marketing bakery products seemed to be more resilient to COVID-induced disruptions. This is not surprising, considering the non-perishable nature of most raw materials, and the local-demand-increasing effect of supermarket closures, or indeed, the limited opening times of large bakeries.

The coefficients associated with factors 1–3 measured the impact of changes in the ranking of specific channel types. The second factor (home delivery, own ecommerce possibilities, and directory) proved to be consistently significant across the models. The positive sign highlights the importance of these marketing channels that best allow for social distancing. The coefficient for Factor 3 (markets, farm gate sales and festivals) is significant only in the full model (Model 3). These channels, although declining somewhat in importance due to the pandemic, remained important, especially farm gate sales.

While Romania is the benchmark, negative and highly significant country dummies suggest that farmers in all other countries had less chance of increasing their sales during COVID. Due to the great diversity of small farms in Romania [14], the optimal production and marketing strategy may have been found by many.

## Conclusions

Small-scale farmers appear to comprise a rather homogenous group in terms of income and thus size; accordingly, economies of scale appear to be a less important factor, unlike economies of scope. Our evidence reveals that diversification as a strategy pays off, both in terms of marketing channels and product categories. However, the impact seems to be nonlinear; the initial advantage of diversification disappears, either temporarily (in the case of products), or permanently (in the case of marketing channels). In other words, an intermediate level of channel diversification appeared to be a successful strategy, while in the case of product diversification, lower and higher levels of diversification paid off. Small producers typically have to manage multiple roles at the same time, thus the selection of the optimal number of marketing channels and products into which to invest their limited time and human resources requires careful consideration. Some diversification increases the chance of selling products for which demand is sustained, or the use of a mode of sales that is successful in a changing situation. Additionally, the results imply that home deliveries and online sales of perishable products suffered. In the latter case, this may be due to the need for more specific forms of delivery infrastructure.

What makes this study an important contribution to the literature is that it focuses partly on the impact of diversification related to marketing channels, about which knowledge is very limited. The relatively large sample (which is unusual in relation to case studies about small-scale farmers and short food supply chains are concerned) described here is analysed using quantitative methods. Further research is needed to verify results not only in other socio-economic contexts, but the sample selection of the current research (e.g. the identification of farmers through their marketing channel use, and cooperation with organizations focusing on

rural development, small-scale farmers and local communities) also poses some limitations. However, the agricultural sectors of the sample countries differ significantly, thus the approach taken here is expected to deliver general and robust outcomes.

## Supporting information

**S1 Table. Country-specific descriptive statistics (Estonia).**
(DOCX)

**S2 Table. Country-specific descriptive statistics (Hungary).**
(DOCX)

**S3 Table. Country-specific descriptive statistics (Portugal).**
(DOCX)

**S4 Table. Country-specific descriptive statistics (Romania).**
(DOCX)

**S1 Data.**
(XLSX)

## Acknowledgments

The authors express their gratitude to the producers who participated in the research, and to Gusztáv Nemes for providing the Hungarian data. The language-related contribution of Simon Milton is gratefully acknowledged. The authors would like to thank the two anonymous reviewers for their constructive comments.

## Author Contributions

**Conceptualization:** Zsófia Benedek, Imre Fertő, Zoltán Bakucs.

**Data curation:** Zsófia Benedek, Cristina Galamba Marreiros, Pâmela Mossmann de Aguiar, Cristina Bianca Pocol, Lukáš Čechura, Anne Põder, Piia Pääso, Zoltán Bakucs.

**Formal analysis:** Zsófia Benedek, Imre Fertő, Zoltán Bakucs.

**Methodology:** Imre Fertő.

**Resources:** Imre Fertő.

**Supervision:** Imre Fertő.

**Visualization:** Zsófia Benedek, Imre Fertő, Zoltán Bakucs.

**Writing – original draft:** Zsófia Benedek, Cristina Galamba Marreiros, Cristina Bianca Pocol, Lukáš Čechura, Anne Põder, Piia Pääso, Zoltán Bakucs.

**Writing – review & editing:** Zsófia Benedek, Imre Fertő, Cristina Galamba Marreiros, Pâmela Mossmann de Aguiar, Cristina Bianca Pocol, Lukáš Čechura, Anne Põder, Piia Pääso, Zoltán Bakucs.

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
