## [Decision Letter · Decision Letter 0]

1 Feb 2021

PONE-D-20-39115

The ideal level of diversification: the economic success of small-scale farmers during the first wave of COVID-19. An international study.

PLOS ONE

Dear Dr. Benedek,

Thank you for submitting your manuscript to PLOS ONE. After careful consideration, we feel that it has merit but does not fully meet PLOS ONE’s publication criteria as it currently stands. Therefore, we invite you to submit a revised version of the manuscript that addresses the points raised during the review process.

We look forward to receiving your revised manuscript.

Kind regards,

Prof. Arkadiusz Piwowar

Wroclaw University of Economics and Business

Academic Editor

PLOS ONE

Journal Requirements:

Reviewers' comments:

Reviewer's Responses to Questions

5. Review Comments to the Author

Reviewer #1: I like the main idea of the article. It is original and contributes to the literature. However, the authors need to correct two basic issues:

- better explain the selection of the research sample and describe the farms they surveyed. Explain what was the method of sample/respondents selection. Why is the sample from Portugal so small? The selection of the sample does not seem reliable and credible. How did you define small-scale farmers? What was a threshold to include a farm in the sample? You write that “Relatively large small-scale farms are overrepresented in the Czech sample (farms with an annual income of 50,000 EUR or more make up about 52% of this specific subsample)”. I am confused not really sure how did you pick up your sample. Are your farms really small-scale farms. Maybe provide more detailed sample statistic. Besides, is your sample anyhow related to FADN? If not, how did you reach these farms and selected them? I suppose that your sample is far from being representative. I am not even sure that it really includes only small-scale farms. In fact, I have no idea what farms you studied? Were they just small, how small, what was their production profile, with the addition of off-farm work, etc?

- figures and tables are unclear and difficult to use/understand. The rule is that the table/figure should be easy to understand even without reading the text. Figures 1 and 2 are rather difficult to understand. Try to present the size structure in different way. Besides, the ranges are overlapping. Table 1 is also unclear. I understand that the column N is the number of farms. First of all, it should be noted that it is the number of holdings in the FADN system and that it is expressed in thousands. Secondly, for each type of production, the number of farms is the same, which obviously is a mistake. Thirdly, in each column to a number of digits after dot should the same. And why are there only zeros in the minimum size of farm? The minimum economic size threshold for a farm to be included in FADN is 8 thousand euro, so it is impossible to have zero ha in case of every country - extend a number of digits after zero. Data for Portugal seems unrealistic. Please double check. Figures 3 and 4 are also not very clear to me. Please describe the axes better.

Some minor issues:

Line 158 – FADN does not use ESU anymore. The economic size of farm is currently measured by standard output (SO). Explain the AWU and how it differs among countries.

Line 233 – You write “The structure of the dataset resulting from the interviews is as follows” but actually not table or information about data structure is following. Just some info how data was collected. This info is partly included in table 3, which is presented later on. But it is not enough.

Line 251 – what do you understand by annual gross income – is it family income, farm income, agricultural income? Per person or total household. With off farm activities included?

Line 338 – what was the estimation method? What kind of regression did you estimated. Was it a logistic regression?

English proof is required. Some minor language issues.

Overall, pay attention to details and try to explain details better.

Reviewer #2: Manuscript number: PONE-D-20-39115

Manuscript title: The ideal level of diversification: the economic success of small-scale farmers during the first wave of COVID-19. An international study

General comments:

The manuscript is very well written and organized. Although the manuscript is not exceptionally evolved in methodological terms or in the results obtained, it discusses a relevant topic in the pandemic context, allowing to bring to light some data obtained through surveys of small farmers.

These two specific contexts (that of the pandemic and that of small farmers) are relevant, the first of which is shrouded in uncertainty and can be characterized by the need for constant reaction to policy measures that change at a breakneck pace, and the second still it is insufficiently studied in Europe, particularly with regard to its importance in the economy and in rural society as a whole. I recommend to the authors some of the recent papers that have been published in the framework of the European SALSA project (mainly that of Rivera et al., I think it may reinforce the discussion and/or the introductory sections a little more) (https://doi.org/10.1016/j.gfs.2020.100395;
https://doi.org/10.1016/j.gfs.2020.100389;
https://doi.org/10.1016/j.gfs.2020.100417;
https://doi.org/10.1016/j.gfs.2020.100416;
https://doi.org/10.1016/j.gfs.2020.100412;
https://doi.org/10.1016/j.gfs.2020.100425;
https://doi.org/10.1016/j.gfs.2020.100427). The sub-section “Specificities of the countries involved in the study” can be also strengthen through the literature. I recommend the paper “Typology and distribution of small farms in Europe: Towards a better picture” (https://doi.org/10.1016/j.landusepol.2018.04.012) and references therein. Therefore, I recommend the manuscript for publication after minor revision.

Detailed suggestions:

L25: Change “As many as 19 percent” to “Approximately 19%” or “About 19%”

L42-43: Add references.

L43: Remove “In order”

L46: Change “On the other hand, lockdown measures meant that” to “Lockdown measures also meant that”

L53: Change “In this paper, the focus is on farmers who” to “In this paper, we focused on the farmers who”

L101: Change “On the other hand” to “Moreover”

L104-105: Please clarify. As you start started the sentence with "In this paper ..." it was not clear whether you refer to your own manuscript or to the paper you just cited in the previous sentence (Hunt et al., 2012)

L134: Remove “in order”

L135: Remove “business”

L149: Remove “Data from countries with very different characteristics were used in the analysis”. This sentence adds nothing.

L233: You refer only Hungary. And the remaining countries?

L255: Remove “in order”

L276: Remove “In order”

L286: Regarding Table 3, it seems to me to make more sense to present the distribution of frequencies for categorical variables than the mean and standard deviation (and by country).

---

## [Author Response · Author response to Decision Letter 0]

15 Mar 2021

A rebuttal letter is included as a separate file that responds to each point raised by the academic editor and reviewers.

---

## [Decision Letter · Decision Letter 1]

1 Apr 2021

PONE-D-20-39115R1

The ideal level of diversification: the economic success of small-scale farmers during the first wave of COVID-19. An international study.

PLOS ONE

Dear Dr. Benedek,

Thank you for submitting your manuscript to PLOS ONE. After careful consideration, we feel that it has merit but does not fully meet PLOS ONE’s publication criteria as it currently stands. Therefore, we invite you to submit a revised version of the manuscript that addresses the points raised during the review process.

We look forward to receiving your revised manuscript.

Kind regards,

Arkadiusz Piwowar

Wroclaw University of Economics and Business

Academic Editor

PLOS ONE

Journal Requirements:

Reviewers' comments:

Reviewer's Responses to Questions

6. Review Comments to the Author

Reviewer #2: Manuscript Number: PONE-D-20-39115R1

Manuscript Title: The ideal level of diversification: the economic success of small-scale farmers during the first wave of COVID-19. An international study.

Regarding the response to the comments I made in the first review, there is nothing to add, the authors complied and responded accordingly. However, I have carefully read the comments of Reviewer # 1, which I consider relevant and important, and therefore deserve special attention by the authors. It is very difficult to achieve good levels of representativeness when the focus of the analysis is on small farms, because there is a lack of knowledge about their real diversity. This fact, in itself, makes the publication of this manuscript important, which relevance is exacerbated by the pandemic context in which we live. Furthermore, the authors never aspired to achieve this representativeness, and this seems clear to me in the manuscript. The issue related to the size of farms can be better explained, and some of my suggestions go in this direction. In my opinion, these issues can be overcome through minor adjustments to the manuscript, so they do not jeopardize its publication, the general objectives, and the analytical process.

Title

I think that the authors should change the title. The authors did not determine the "ideal level", for that you would need other statistical procedures, and a larger and more diverse sample. I think it would be more adjusted to the manuscript content to reformulate the first part of the title as a question. The second part of the title (“An international study”), in addition to being unappealing, creates expectations in the reader that can then be frustrated. Thus, my proposal is:

Is farm diversification a success factor for small-scale farmers constrained by COVID-related lockdown? Contributions from a survey conducted in four European countries during the first wave of COVID-19

Introduction (and related 3 sub-sections)

L48-49: Suggestion: Change “The virus, simultaneously causing global and local economic as well as social disturbances [11], (…)” to “The virus, while simultaneously causing economic and social disturbances at multiple scales [11], (…)”

L51: Move the sentence "Small farms have been (...)" to a new paragraph

L68: Consider change the title of the sub-section to “Linkages between Short Food Supply Chains and small-scale farmers”

L69-70: Simply stating that the interest remains “undiminished” is not enlightening. The reader will remain unsure whether the interest is low or high.

L70-72: Reformulate the sentence. Suggestion: “Despite the growing attention of researchers and policy makers to local food systems, alternative food networks and short food supply chains, their respective definitions remains unclear.”

L72-75: Change the sentence as follows (suggestion): “In our study we followed the approach of Gruchmann et al. [18] and Schmutz et al. [17], mainly focusing on producer-consumer interactions involving producers directly selling their products to consumers, or through a limited number (ideally, zero) of intermediaries.”

L78: Change “(…) many pieces of national legislation [19] (…)” to “(…) several legislative instruments [e.g., 19] (…)”

L87-93: Rephrase as follows (suggestion): “As above mentioned for short food supply chains, also defining small-scale farmers is challenging [14, 23] and it is often based on certain thresholds that are highly dependent on the geographical context of the analysis [14]. Since the countries involved in our study represent markedly different contexts [14, 24] and building on the considerations of Kneafsey et al. [21] and Martinez et al. [22], small-scale farmers were identified through their participation in short food supply chains, instead of using a specific threshold.”

L93-96: Move this sentence to the Material and Methods section.

L161-251: The subsection "Specificities of the countries involved in the study" should be moved to the Results’ section. The authors explore statistical data that characterize the target-countries of this manuscript, and discuss their differences and specificities. It does not seem to me an introductory section built on the basis of the state of the art, but results. If this option is followed by the authors, they should separate the methodological elements and integrate them in the Materials and Methods section.

L162-163: Remove the first sentence. The idea is repeated in the third sentence of this paragraph.

L177-179: Remove parentheses

L200-201: Remove “In an excellent review,“ and change “finds” to “observed” (or to “found”, in alternative).

L210: Change “The average size of a cereal farm is greatest in Estonia, where the latter are 20 times larger than their Portuguese counterparts” to “The largest mean cereal farm size can be found in Estonia, about 20 times larger than the Portuguese value, the lowest one, according to the data from FADN”. Additionally, at first reading the distribution of this values (Table 1) seems awkward, because they seem to contradict the majority of scientific publications on these topics. However, the authors must reinforce that this values are related to “production mix” farms, and clarify and discuss the rationale behind this option (most likely, this nuance was not clear to Reviewer #1, which led him to question the reliability of this data).

L211: Change “picture” to “distribution”

L231: Remove “banned”

L233-236: Rephrase as follows: “Telework was centrally mandated for some professionals, such as civil servants in Portugal, but many more decided to stay at home in all countries, either to supervise their children (as the institutions of education had closed everywhere), making the use of the home office as a general rule.”

L236: Change “Though” to another synonim to avoid two sentences in the same paragraph starting with the same word.

L255: Change “help” to “contribution”

Material and methods

L262: Move the sentence from L93-96 to this point. Additionally, for non-European readers, the authors must clarify what the LEADER Local Action Groups are, and their relation with small farming, so that there is no doubt that your sample is made up of small farmers. This doubt of Reviewer #1 seems to me not to be completely clarified in this version of the manuscript.

L263: Move to another paragraph

L272-273: Change as follows: “Concerning the product categories commercialized by each farmer, data were coded into dummies: “1” if yes, “0” otherwise.”

L282: Change “calculations” to “estimations”

L302: Change “explanatories” to “explanatory variables”

L305-306: Remove “Thus, the straightforward option for model estimation would be a logit or probit model.” You did not use any of these, why mention?

L306-309: Rephrase as follows: We used the semi-nonparametric (SNP) method defined by Gallant and Nychka [54] due to its robustness when compared to the standard models.” and I think you can remove this part “(…) – furthermore, a post-estimation test of the null hypothesis whether a probit estimation would suffice, was considered easy to conduct.”

L310-311: Change “(…) non-parametric approach is that, unlike parametric estimators, it is not sensitive to departures from distributional assumptions (…)” to “non-parametric approach is that, unlike parametric estimators, it is not sensitive to distributional assumptions (….)”.

Results and discussion

L334: Change “earlier” to “previous”

L335: Change “tended” to “tend” and “market” to “sell”

L336-338: Reformulate this sentence to increase readability

L365: Change “appeared” to “seems”

L390: I think you need to add an additional measure of performance of each model (an adjusted R2; or the deviance D2).

---

## [Author Response · Author response to Decision Letter 1]

7 Apr 2021

A rebuttal letter is included as a separate file that responds to each point raised by the Reviewer.

---

## [Decision Letter · Decision Letter 2]

3 May 2021

Farm diversification as a potential success factor for small-scale farmers constrained by COVID-related lockdown. Contributions from a survey conducted in four European countries during the first wave of COVID-19.

PONE-D-20-39115R2

Dear Dr. Benedek,

We’re pleased to inform you that your manuscript has been judged scientifically suitable for publication and will be formally accepted for publication once it meets all outstanding technical requirements.

Kind regards,

Arkadiusz Piwowar

Wroclaw University of Economics and Business

Academic Editor

PLOS ONE

Reviewers' comments:

Reviewer's Responses to Questions

6. Review Comments to the Author

Reviewer #1: I like the changes made to the title and other enhancements made under the influence of the second reviewer.

However, I do not understand why the authors ignore my comments about the need to standardize the writing of numbers in tables. I informed about it in both reviews. This is just a technical question. Maybe they don't understand what the problem is. So I give an example. In Table 1 Min for Estonia is 0.1 and for Portugal it is 0.01. For the record to be consistent, 0.10 should be written for Estonia. If the variable was written for one country with two decimal places, the data for other countries should also be presented in the same way. This note applies to all tables.

Reviewer #2: Manuscript Number: PONE-D-20-39115R2

Manuscript Title: Farm diversification as a potential success factor for small-scale farmers constrained by COVID-related lockdown. Contributions from a survey conducted in four European countries during the first wave of COVID-19

General comments:

The authors once again did a remarkable job of adapting the manuscript to the reviewers' suggestions. The manuscript maintains its interest and has increased readability. It is ready to be published by PLOS ONE. I wish the best of luck to the authors in their future publications.

---

## [Editor Report · Acceptance letter]

14 May 2021

PONE-D-20-39115R2 

Farm diversification as a potential success factor for small-scale farmers constrained by COVID-related lockdown. Contributions from a survey conducted in four European countries during the first wave of COVID-19. 

Dear Dr. Benedek:

I'm pleased to inform you that your manuscript has been deemed suitable for publication in PLOS ONE. Congratulations! Your manuscript is now with our production department. 

Kind regards, 

on behalf of

Professor Arkadiusz Piwowar 

Academic Editor

PLOS ONE